# Study on Generating Machining Performance of Two-Dimensional Ultrasonic Vibration-Composited Electrolysis/Electro-Discharge Technology for MMCs

**DOI:** 10.3390/ma15020617

**Published:** 2022-01-14

**Authors:** Jing Li, Wanwan Chen, Yongwei Zhu

**Affiliations:** 1School of Mechanical Engineering, Yangzhou University, Yangzhou 225127, China; dx120180056@stu.yzu.edu.cn (J.L.); dx120190080@stu.yzu.edu.cn (W.C.); 2College of Hydraulic Science and Engineering, Yangzhou University, Yangzhou 225127, China

**Keywords:** 2D ultrasonic vibration, electrolysis/discharge, MMCs, MRR, surface quality, form precision

## Abstract

Ultrasonic vibration-composited electrolysis/electro-discharge machining technology (UE/DM) is effective for machining particulate-reinforced metal matrix composites (MMCs). However, the vibration of the tool or workpiece suitable for holes limits the application of UE/DM. To improve the generating machining efficiency and quality of flat and curved surfaces, in this study, we implemented two-dimensional ultrasonic vibration into UE/DM and constructed a novel method named two-dimensional ultrasonic vibration-composited electrolysis/electro-discharge machining (2UE/DM). The influence of vibration on the performance of 2UE/DM compared to other process technologies was studied, and an orthogonal experiment was designed to optimize the parameters. The results indicated that the materiel remove rate (MRR) mainly increased via voltage and tool vibration. The change current was responsible for the MRR in the process. Spindle speed and workpiece vibration were not dominant factors affecting the MRR; the spindle speed and tool and workpiece vibration, which reduced the height difference between a ridge and crater caused by abrasive grinding, were responsible for surface roughness (Ra) and form precision (δ). Additionally, the optimized parameters of 1000 rpm, 3 V, and 5 um were conducted on MMCs of 40 SiCp/Al and achieved the maximum MRR and minimum Ra and δ of 0.76 mm^3^/min, 3.35 um, and 5.84%, respectively. This study’s findings provide valuable process parameters for improving machining efficiency and quality for MMCs of 2UE/DM.

## 1. Introduction

MMCs have outstanding advantages, but the performance of the two constituent materials differs. Therefore, conventional Grinding machining (GM) or special machining (such as electrolysis machining (ECM) and electro-discharge machining (EDM)) usually cannot achieve satisfactory machining efficiency and quality at the same time and tool wear is severe [1,2,3]. GM causes cracking, splitting, particle displacement, and serious tool wear [4,5]. Electric discharge machining (EDM) produces a recast layer by melting, enhances the shielding effect of particles, and reduces the machining efficiency [6]. In recent years, electrolysis/electro-discharge machining under a low current density has become the most promising technology for solving the problems experienced when machining composites. Electrolytic discharge machining (ECDM) is a technology that applies bubbles generated via electrolysis to form a gas film around electrodes and then induces discharge melting and chemical corrosion between a tool and a workpiece [7]. The nonconductive material in ECDM is mainly removed by the discharge between the gas film and workpiece, with chemical etching as the auxiliary method [8]. Although electrolytic/electro-discharge machining (E/DM) uses electrolysis and discharge phenomena, it is different from ECDM [9]. E/DM combines the characteristics and advantages of ECM to remove metal material; EDM between a tool and workpiece and ECDM between a gas film and workpiece remove nonconductive materials as auxiliary methods. High-temperature melting and heat-accelerated chemical etching are applied on composite materials, and electrolytic dissolution of pits and protrusions is applied to reduce surface roughness. Liu [10] used GE/DM to machine MMCs and found that the removal mechanism is similar to that of ECDM and grinding, compounded to eliminate the recast layer and reduce the surface roughness by a diamond, abrasive, particles-coated tool. Kanagaraja [11] found that when the mass fraction of reinforced particles was higher, the cutting force needed to shear or break ceramic particles was greater. When the rotational speed of the tool electrode was increased, it had positive effects on machining performance.

The preceding analysis shows that the removal rate and machining precision of MMCs need to be enhanced. Many researchers have reported that a high vibration frequency or high vibration amplitude enhances electrolytic machining performance characteristics [1,2,3,4,5,6,7,8,9,10,11,12,13]. Anasane [14] and Ghoshal [15] reported that increasing tool vibration amplitude improves the performance characteristics and gap environment of ECM. Pawariya [16] used ultrasonic vibration of tool electrode of ECDM to investigate and enhance deep microholes in difficult-to-machine metal materials. Wang [17] reported that tool vibration improves machining the efficiency, surface quality, and gap environment of ECM.

The abovementioned machining methods are mostly suitable for the microhole machining of metal, conductive materials. For nonconductive materials, ultrasonic vibration-composited electro-discharge machining is a suitable choice [18]. Razfar [19] and Elhami [20] introduced vibration to a tool to machine glass microholes and found that ultrasonic vibration increased the frequency and probability of discharges and improved the material removal rate and surface quality [21]. Schubert [22] found the processing speed can be increased by 40% and the aspect ratio of metal micromoles can reach 40.

The above ultrasonic vibration-composited electro-discharge and electrolytic machining have outstanding advantages in applications with difficult-to-machine materials, such as enhancing machining efficiency and precision, intensifying the renewal of electrolyte, or reducing tool wear [23]. For MMCs that are difficult to machine using conventional or single unconventional processes, ultrasonic vibration is effective for machining reinforced matrices with higher hardness. However, thus far, there is no report on the generating machining of flat or curved surfaces using 2UE/DM.

Combining the advantages of the high surface quality produced by ultrasonic vibration of EC/DM and the high efficiency by grinding on MMCs, we constructed a method called two-dimensional ultrasonic vibration-composited electrolysis/electro-discharge (2UE/DM) that functions under low current density and low voltage to improve the machining efficiency and quality of MMCs. During the processing, the ultrasonic vibration direction of the tool and workpiece are axial (Z direction) and tangential (X direction), respectively. A schematic diagram of side-generating machining on a surface by 2UE/DM is shown in Figure 1. In this study, we analyzed the material removal mechanism of 2UE/DM using comparative experiments on SiCp/Al with different processes, and examined the material remove rate (MRR), surface roughness (Ra), and form precision (δ) through experiments with different process parameters.

## 2. UE/DM Machining Mechanism

The proposed 2UE/DM is a technology combining two-dimensional ultrasonic vibration-assisted grinding machining (2UM), electrolysis machining (ECM), and discharge machining (DM). The gas and fluid in the machining gap are uniform and consistent under the effect of two-dimensional ultrasonic vibration, and the change in electrolyte conductivity by concentration difference, heat, and gas can be ignored. A conductive tool coated with diamonds was connected to the anode of the power source, whereas the workpiece was connected to the cathode of the power source; the gap between the electrodes was filled with passive electrolyte with low conductivity. The applied voltage was below 10 V and the current density was about 5–15 A/cm^2^. The tool rotated, fed along the X direction, and vibrated axially simultaneously (with an amplitude of A_Z_ and a frequency of f_Z_). The workpiece vibrated tangentially along the tool feed direction (with an amplitude of A_X_ and a vibration frequency of f_X_). The surface material of the workpiece was removed by a different removal mechanism with diffident displacements of ultrasonic vibration of the workpiece, as shown in Figure 2. In the machining process, the electrolytic dissolution rate was mainly affected by A_X_ and the initial machining gap of G(*t*_0_). The axial vibration and rotational movement of the tool had a slight effect on the machining gap. During one feed cycle of the workpiece, the machining gap G(*t*) mainly varied periodically with the vibration of the workpiece, which can be expressed by Equation (1).
(1)$$G\left(t\right)=\;G\left(t_{0}\right)+A_{X}\sin\left(2{\pi f}_{X}t\right)$$

According to Faraday’s law of electrolysis, the current density i mainly depends on the applied voltage drop U_R_ and the machining gap G(*t*), $\;i\sim \frac{U_{R}}{G\left(t\right)}$. The vibration of the workpiece leads to the largest machining gap at *t*_1_; additionally, the current density between the electrodes is lower, which avoids a short circuit or blockage occurring. The machined products and heat are ejected via pumping and renewing the electrolyte; this improves and stabilizes the machining environment [24]. The vibration of the workpiece leads to the smallest machining gap at *t*_3_, when the electrolytic dissolution rate of the metal matrix reaches a maximum. Additionally, abrasive particles grind the workpiece and remove material; this achieves higher MRR and localized machining ability under low voltage of ECM.

The DM means two types of machining: ECDM and EDM. The bubbles generated from the two electrodes accumulate on the surface of the tool under the action of buoyancy and surface tension; then, a gas film forms [25]. As the vibration of a workpiece increases, the machining gap G(*t*) becomes larger than the thickness of the gas film (the thickness of the gas film is generally 0.1–0.2 mm [24]); thus, the gas film isolates the tool and electrolyte, and ECDM occurs between the workpiece and electrolyte. If G(*t*) is less than the thickness of the gas film, the gas film insulates the electrodes. When the gap between the electrodes is 10–20 μm lower than the discharge gap, EDM occurs between the tool and workpiece, as shown in Figure 2. Although a static electrolyte is conducive to the formation of a more stable insulating gas film, an electrolyte or tool with compounded ultrasonic vibration is more conducive to the formation of a gas film and to the reduction in the thickness of the gas film; therefore, electro-discharge machining is possible under lower voltage [13]. The initial state between the electrodes is restored after electro-discharge machining and deionization. The distance between electrodes when the vibration of a workpiece is varied allows the electric field between the electrodes to still reach the discharge conditions and increases the probability and frequency of discharge, effectively removing passivation bumps in electrolysis to enhance the MRR.

The abrasive particles on the tool surface grinding machines in the ellipse trajectory of 2UM are similar to those of tangential microgrinding machining on the surface of the workpiece. In each microgrinding process, abrasive particles exhibit three types of machining. (1) The tangential ultrasonic impact on a workpiece causes abrasive particles to periodically hammer the surface of the workpiece, thereby obtaining smaller debris and varying the cutting depth of each microgrinding process. (2) In the microgrinding process, the abrasive particles are similar to the microgrinding blade and abrade the workpiece to produce tiny debris. The axial vibration of the tool extends the cutting arc length of the abrasive particles, enhances the removal ratio in the ductile zones of reinforced particulates, and widens the width of the grooves scraped by abrasive particles. (3) Under the joint action of 2D vibration and rotary grinding, rolling- and milling-like grinding are performed on the surface of the workpiece with wider coverage; this reduces protrusions after grinding and electrolytically insoluble points and strengthens the fractures and bulges of reinforced particles [10]. Hammering and abrasion of reinforced particulates is the major material removal form in 2UM, whereas the rolling action varies the surface morphology and improves the surface quality [26].

## 3. Experimental Setup

Figure 3a shows a schematic diagram of the 2UE/DM experimental device. Figure 3b shows the in-house-built experimental equipment. The equipment was an ordinary grinder that included a Z-direction ultrasonic vibration head modified from a BT30 configured with a slip ring. During machining, the current was transmitted to the tool through a slip ring insulated from the machine tool and Z-direction ultrasonic generator. The workpiece was clamped on an elastic support frame that could slide in the X direction, and the vibration was transmitted to the workpiece through an X-direction ultrasonic generator. The vibration frequencies of the tool and workpiece were 18.33 and 19.46 kHz, respectively.

The workpiece was particulate-reinforced aluminum alloy 6061 with 40 vol.% SiC reinforcement (40 SiCp/Al), as described in Table 1. The tool material was WC with a diameter of 6 mm coated with diamond abrasive particles (100#). The electrolyte was a 0.5 wt.% NaCO_3_ solution pumped into the machining area through a nozzle during machining. The initial work piece was 50 × 50 × 5 mm, and the machining surface was 50 × 5 mm ground and cleaned before machining.

To compare the forming quality and machining efficiency of 2UE/DM to other processes and to study the specific effects of 2D ultrasonic vibration, comparative tests were designed for clarifying the mechanisms: general grinding machining (GM), without vibration (GE/DM), Z vibration (ZUE/DM), X vibration (XUE/DM), and 2UE/DM, as shown in Table 2. From the four sets of signal factors tests, the effects of different process parameters on the MRR, surface quality, and edge precision of 2UE/DM were analyzed. A comparison of the parameters is shown in Table 3, and other parameters are shown in Table 2.

The power of the ultrasonic generator could be adjusted to vary the amplitudes of the workpiece and tool, and a laser microdisplacement sensor was used to measure the amplitude before the machining test. The machining time of each test was 3 min. A balance that had a precision of 0.1 g was used to measure the removed mass and to calculate the MRR. Each test was performed three times, and the average value and standard deviation were calculated. The surface roughness of the machined surface was measured to evaluate the surface quality. A stereo-measuring microscope was used to observe the micropattern, and a Countour GT-K 3D optical profiler was used to measure and image the Ra and 3D appearance of the workpiece. The form precision [27] can be evaluated by the ratio of the cross-sectional area S_b_ of the notch in the surface actually machined to the theoretical cross-sectional area S_c_, as shown in Figure 4. L denotes the measured length, and H denotes the depth of the largest notch. Then, the form precision δ can be expressed by:(2)$$\delta =\frac{S_{b}}{S_{c}}=\frac{S_{b}}{\mathrm{LH}}$$

## 4. Comparative Analysis of Different Machining Mechanisms

Figure 5 shows the waveforms of different machining currents. The gap between the tool and workpiece does not change theoretically in GE/DM and ZUE/DM, while the electrolysis current (CH2: I = *i*S, where S is the machining area) of ZUE/DM is almost the same without an ultrasonic wave, shown by the electrolytic/electro-discharge current (CH1) in Figure 5a,b. However, the electrolyte in the interelectrode gap cannot be renewed fast enough, the current stability of GE/DM is less than that of ZUE/DM, and discharges are rare in both; this is consistent with the analysis of the machining mechanism. In Figure 5c,d, the gap $G\left(t\right)=G\left(t_{0}\right){+A}_{X}\sin\left({2\pi f}_{x}t\right)$ changes periodically with the vibration of the workpiece. The electrolysis current of 2UE/DM and ZUE/DM varied with the periodic vibration amplitude, a sudden increase due to EDM or ECDM occurred in the gap, and the frequency of discharges in 2UE/DM was significantly higher than that in ZUE/DM. This finding shows that two-dimensional ultrasonic vibration helps to increase the probability and frequency of discharge. It also avoids discharge at the same point and prevents the occurrences of short-circuit, arcing, and other phenomena. Electrolysis is an important method of removing material, and electro-discharge is more conducive to removing hard-to-dissolve and the reinforced fractured bumps of ceramic-based particles; they enhance the efficiency and stability of electrolytic machining.

Figure 6 shows the MRR data based on the experimental results of five machining processes. The varying machining gap helped to pump and renew the working fluid and discharge the machined products. The MRR of XUE/DM was slightly higher than that of ZUE/DM. Under the composited effect of 2D ultrasonic vibrations, the varied gap improved the machining efficiency. The MRR of 2UE/DM was higher by 18.9%, 13.1%, 7.8%, and 9.5% than that of GM, GE/DM, XUE/DM, and ZUE/DM, respectively. When there was no vibration, the uncycled electrolyte deteriorated the environment of machining gap gathered a large amount of material removal products and bubbles and reduced the material removal volume of E/DM. The abrasive particles not only scraped and activated the surface of the workpiece, but they also had a deeper grinding depth than that of GM, GE/DM, and ZUE/DM because of the X-direction vibration. This indicated that impact force in larger ultrasonic amplitude acted on the depth of wear debris, so it had a greater probability of the ceramic particles to be scraped or pulled out along the X direction, thereby significantly enhancing the machining efficiency.

Different motion trajectories of the abrasive particles with two-dimensional ultrasonic vibration simultaneously improved the surface quality of grinding and electrolysis/electro-discharge machining on the surface of the workpiece. The surface roughness and three-dimensional morphologies are shown in Figure 7 and Figure 8, respectively. As seen from the morphology formed by GM, there were obvious shapes of ridges and craters; these were cutting marks made by the abrasive particles. The maximum height difference between the ridges and craters was relatively large (about 155 μm). Pits and fracture marks of ceramic particles were visible at the material interfaces, showing a poorer surface quality; additionally, Ra was as high as 11 μm. In Figure 8b, the craters formed by GE/DM abrasive particles were shallower than those formed by GM with a depth of 147 μm; Ra decreased to about 6 μm due to the electrolytic machining reducing the maximum height difference. The morphology formed by ZE/DM is shown in Figure 8c. The axial vibration increased the proportion of the plastic area, and only a small amount of debris breakage occurred when the surface was ground. The mechanism can basically be regarded as plastic shearing removing the ceramic particles. When the metal matrix dissolved, the squeezed zone and the marks of plastic grinding were removed, including the crater zone, while some particles were exposed. The ridge zone showed a small amount of residual fracture under the action of electrolysis and vibration. The maximum height difference decreased to 130 μm, and the ridge-crater morphology was not as obvious as in GE/DM and GM. This indicated the increased smoothness of the machined surface, with Ra being 20% lower than for GE/DM. Figure 8d shows that the cutting depth of abrasive diamond particles in XUE/DM was deeper than GM; only a few brittle fractures were observed on the surface. Although the impact force of A_X_ increased the multilevel microcracks on ceramic particles, the exposed broken particles by ECM were pulled up and removed plastically. The repeatedly rolled area on the grinding surface increased, while the residual height decreased significantly along the tangential direction, and the surface height difference was only about 139 μm. Figure 8e shows that the surface formed by 2UE/DM was mainly composed of a grinding zone of plastic particles and an electrolysis zone of a metal matrix. Electrolysis of the metal matrix led to a reduction in the regular traces of machined plastic craters and ridges under the action of two-dimensional vibration. The proportion of the plastic machining zone was relatively large, and a single particle broke and fell off because of the dissolution of the metal matrix. The variations in morphology showed that the two-dimensional vibration and the rapid dissolution of the metal matrix accelerated the removal of composites. This was associated with the axial vibration plastic domain grinding of tools and the feed direction of vibration rolling. Thus, the maximum height difference between ridges and craters decreased by 125 μm, and Ra decreased to 3.4 μm. The surface of the workpiece was more flat and uniform, and the surface quality was significantly improved.

As observed from the microtopography of the machined edge (Figure 9), the form precision differed by different machining processes. More particles fell off the edge and more edge notches occurred in GM due to the dissolving of the metal matrix, as shown in Figure 9b. A higher proportion of plastic zones made the edge relatively flat with ZUE/DM. The edge notches were filled with the metal matrix in XUE/DM and 2UE/DM after the exposed broken ceramic particles were dragged and crushed. However, the latter had fewer notched and neat edges. These topographic characteristics verified the removal mechanism of MMCs in 2UE/DM and its efficacy. As shown in Figure 10, the precision of the edge produced by 2UE/DM was the lowest (only about 6.8%) and slightly lower than that produced by ZUE/DM and XUE/DM; however, it produced reductions of about 24% and 47% compared to GE/DM and GM, respectively. The results indicated that the two-dimensional ultrasonic vibration increased MRR while enhancing the machining quality. The machining performance of 2UE/DM was significantly better than that of the other four machining techniques: 2UE/DM is a high-speed, high-quality, and reliable technology for machining MMCs.

## 5. Effects of Process Parameters on 2UE/DM

### 5.1. MRR

Figure 11 shows the MRR of 2UE/DM of different process parameters. The MRR did not increase with an increase in the rotational speed, and MRR decreased rapidly at 5000 rpm. The gathered bubbles in the gap did not easily form a gas film under the action of centrifugal force, reducing the discharge frequency of DM and the dissolution rate of ECM. Voltage had a stronger impact on the MRR, while the MRR reached 0.89 mm^3^/min in 6 V, indicating that, with an increase in voltage, the electrolysis/electro-discharge action between electrodes was intensified and the percent of volume removed by electrolysis/electro-discharge increased with an increase in voltage [26]. A_X_ had a significant impact on the MRR instead of A_Z_. Grinding depth was greater and the volume of material removed by the grinding process was larger with an increasing A_X_. The MRR at 5 μm was 1.3 times that at 2 μm. The variation in the gap was beneficial for the formation of discharge channels, the cycles of electrolyte, optimization of the machining environment, and enhancement of the MRR. Therefore, to obtain a higher MRR for composites, lower rotational speed, higher voltage, and higher two-dimensional amplitude are recommended.

### 5.2. Surface Quality

The effects of 2UE/DM under different process parameters on surface roughness are shown in Figure 12. As the rotational speed increased, the number of particles that participate in grinding per unit time correspondingly increased and the surface roughness slightly decreased [28]. The depth and rate of the erosion on the metal matrix were greater with higher voltage, while more bubbles generated via electrolysis and electro-discharge between electrodes were more likely to occur and form pits. Then, reinforced particles were more likely to be exposed, resulting in a significant increase of Ra. The surface roughness at 6 V was found to be 1.4 μm higher than at 3 V. The abrasive particles increased the distance between the ridges as A_Z_ increased and reduced the residual height as A_X_ increased, resulting in lower Ra. The A_X_ was more advantageous than that of A_Z_ for improving surface quality. Electrical machining can reduce the actual chip depth of single grinding and accelerate material removal when the gap is small; in particular, it can reduce the residual height after cutting, thereby improving the surface quality after machining [29]. A reduction in cutting depth is beneficial for reducing surface damage. Therefore, to obtain a higher surface quality for a composite material, a lower speed and voltage and a larger two-dimensional ultrasonic amplitude are options to consider.

### 5.3. Form Precision

Figure 13 shows the effects of different parameters on the edge precision in 2UE/DM. As seen, a higher rotational speed of the tool, greater than 3000 rpm, was more likely to cause more edge notches and decrease the form precision. When the applied voltage increased to 6 V, δ increased accordingly because of the rapid dissolution of the metal matrix under a higher voltage. Additionally, the ceramic particles exposing and falling off under the action of abrasive particles led to poorer edge precision. Edge protrusions were more prone to inducing discharge with higher voltage, and then the cross-sectional area of the notch increased. The increase of A_X_ and A_Z_ facilitated the alternating action of abrasive particles on the edge, reducing the grinding force and simultaneously increasing the rolling-covered zone. When the amplitude was 5 μm, the form precision was reduced by about 27% and 21% compared to 2 μm. To obtain higher forming quality, a smaller applied voltage, an appropriate rotational speed, and a larger amplitude may be selected.

### 5.4. Optimal Experiments

In the single-factor tests in the previous section, the comprehensive influence of process parameters on machining performance could not be determined. Therefore, it was necessary to obtain the optimal process parameter design through orthogonal analysis. We considered spindle speed, voltage, and the vibration amplitude of the workpiece and tool as influential factors. Each factor had two levels (Table 4). A Taguchi test design (L_8_) was established using Minitab software to optimize MRR, RA, and δ [30]. Table 5 shows that the maximum MRR occurred at 5000 rpm, 6 V, and 5 μm, whereas the minimum Ra and δ occurred at 1000 rpm and 3 V. The impact of different factors followed the sequence of U > A_X_ > A_Z_ > n. Table 6 shows that U and A_X_ always exhibited a positive effect on the MRR, Ra, and δ, whereas A_Z_ and n had no significant impact on the MRR and δ individually. The result is reasonable since the vibration enhanced the EC/DM action and increased the cutting depth, reducing the height difference [31]. Verification experiments using the optimizing parameters (1000 rpm, 3 V, and 5 μm) were conducted on 40 SiCp/Al, and the MRR, Ra, and δ were 0.76 mm^3^/min, 3.35 μm, and 5.84%, respectively. An enlarged view of the machined surface and edge is shown in Figure 13 and Figure 14, which indicate that 2UEC/DM showed higher machining efficiency and forming quality in machining MMCs.

## 6. Conclusions

In this research, the 2UE/DM method was developed for machining hard and brittle conductive materials of MMCs (40 SiCp/Al). Our experimental research focused on the MRR, Ra, and δ under different machining processes and parameters. According to the test results, we drew the following conclusions:(1)Due to the addition of two-dimensional ultrasonic vibration, the removal mechanism of MMCs in generating machining of EC/DM changed. The 2UE/DM combined the advantages of 2UM, ECM, and DM, improving the machining efficiency and quality significantly.(2)Despite the higher dissolution rate of the metal matrix in 2UEC/DM enlarging the height difference between ridges and craters, the two-dimensional ultrasonic vibration extended the abrasive cutting path and scraped off the exposed ceramic matrix particles and hard-to-dissolve bumps plastically, resulting in a lower surface roughness.(3)The machining performance correspondingly increased with higher process parameters except for the spindle speed, which easily caused edge chipping of the machined surface and increased the form precision. The Z amplitude had no significant impact on the MRR. All the selected parameters of the X amplitude and applied voltage had a considerable impact on the MRR, surface quality, and form precision.(4)The optimized process parameters for 40 SiCp/Al included a spindle speed of 1000 rpm, a voltage of 3 V, and two amplitudes of 5 μm, achieving a maximum MRR of 0.76 mm3/min and minimum Ra and δ values of 3.35 μm and 5.84%, respectively.

## Figures and Tables

**Figure 1 materials-15-00617-f001:**
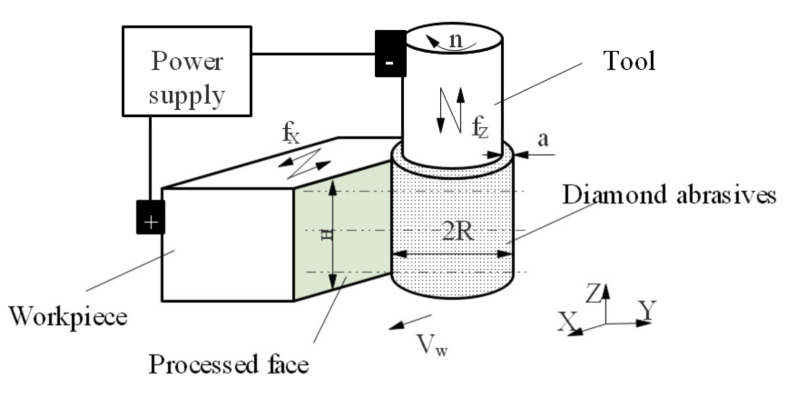
Schematic diagram of side-generating machining of 2UE/DM.

**Figure 2 materials-15-00617-f002:**
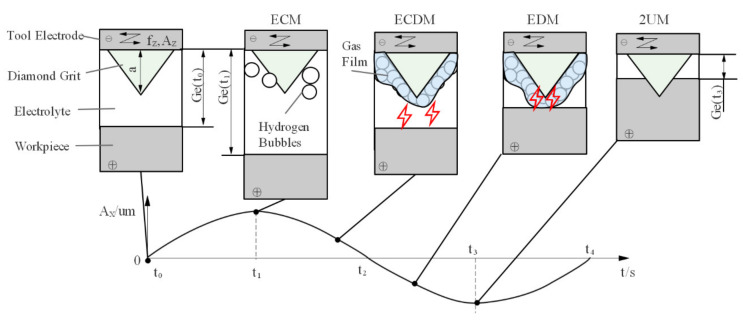
Schematic diagram of the 2UE/DM mechanism.

**Figure 3 materials-15-00617-f003:**
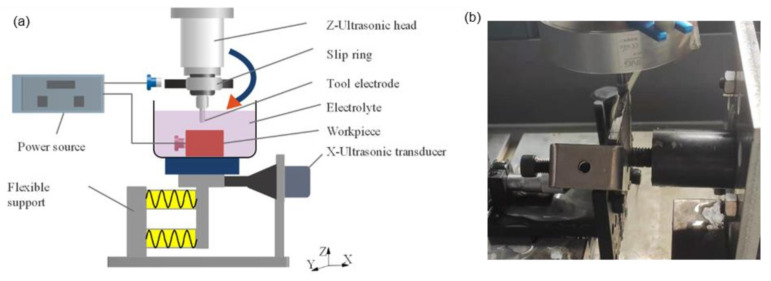
(**a**) Schematic diagram of the experimental setup and (**b**) the actual in-house-built equipment.

**Figure 4 materials-15-00617-f004:**
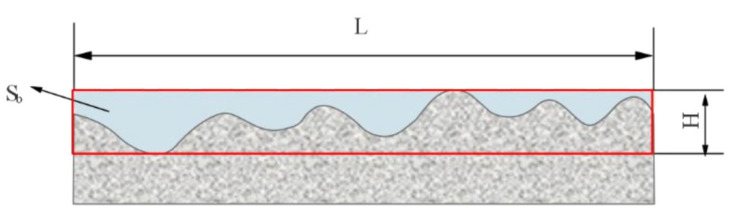
Method for calculating form precision.

**Figure 5 materials-15-00617-f005:**
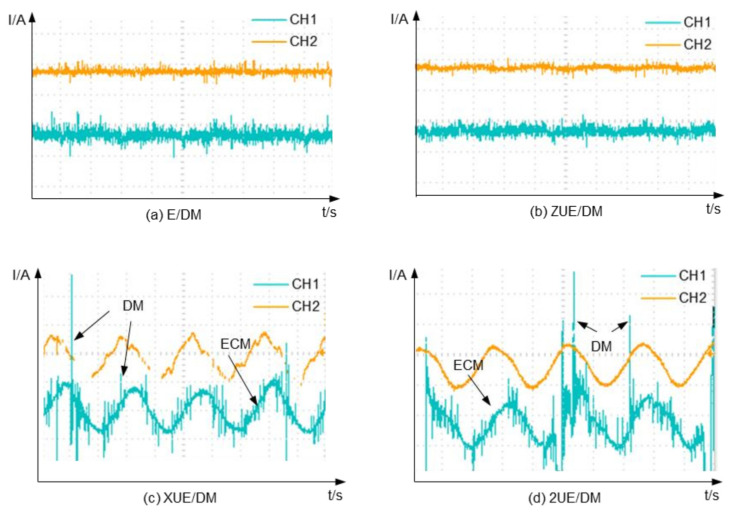
Comparison of current signals: (**a**) GE/DM; (**b**) ZUE/DM; (**c**) XUE/DM; (**d**) 2UE/DM.

**Figure 6 materials-15-00617-f006:**
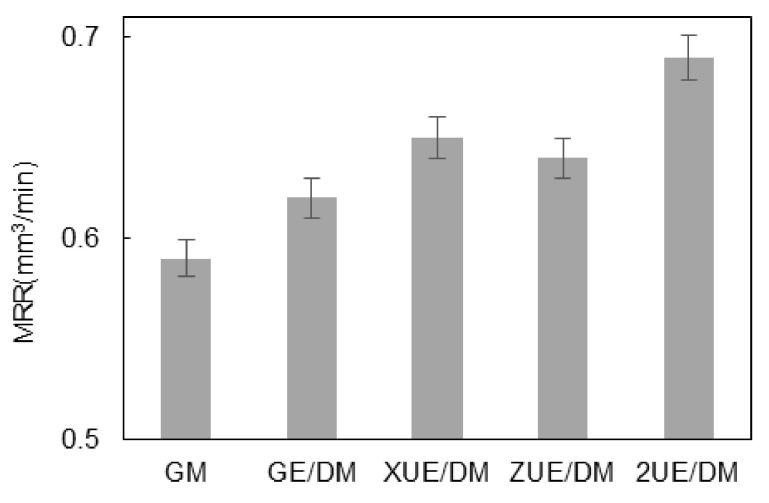
Comparison of machining efficiencies.

**Figure 7 materials-15-00617-f007:**
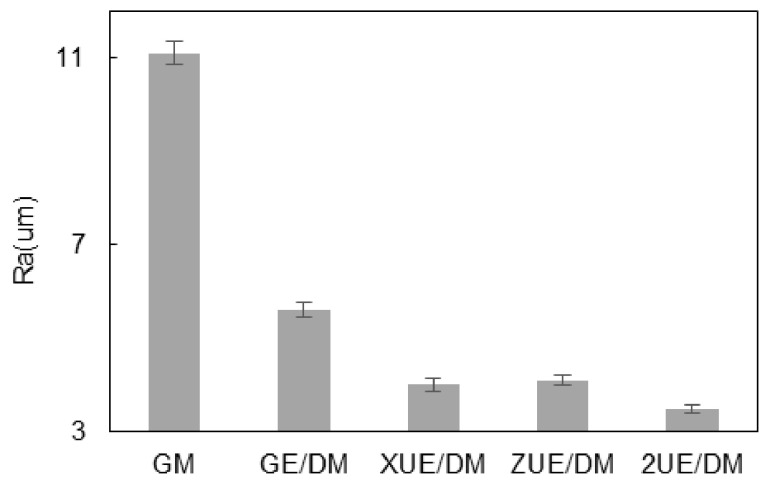
Comparison of surface roughness.

**Figure 8 materials-15-00617-f008:**
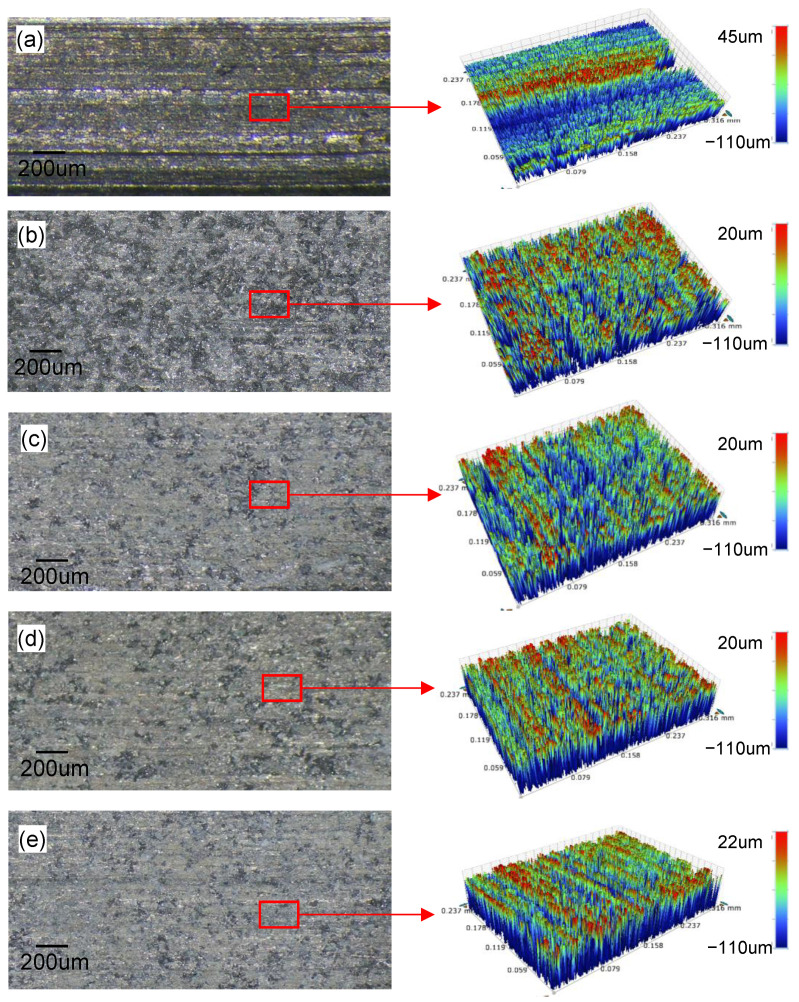
Comparison of machined surface and topographies: (**a**) GM; (**b**) GE/DM; (**c**) ZUE/DM; (**d**) XUE/DM; (**e**) 2UE/DM.

**Figure 9 materials-15-00617-f009:**
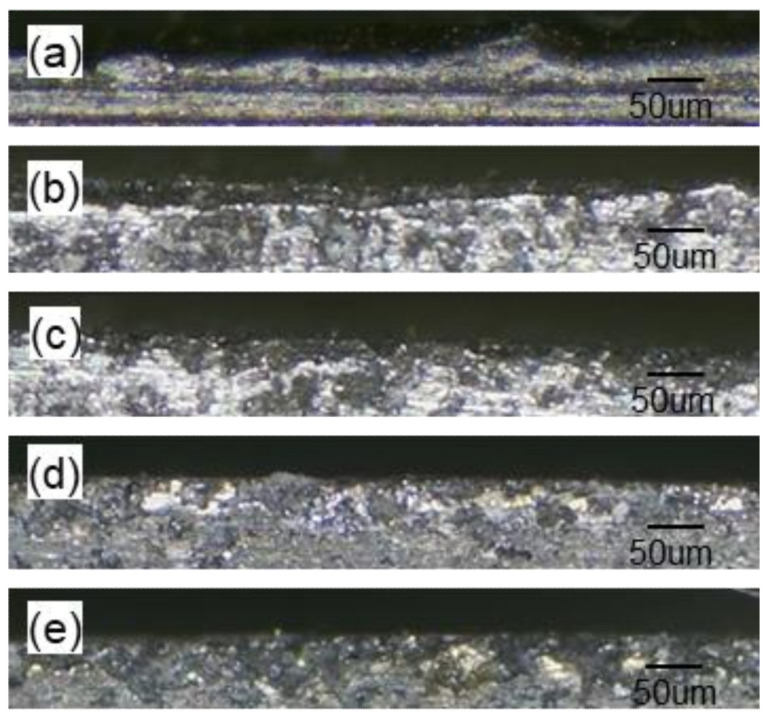
Comparison of edges: (**a**) GM; (**b**) GE/DM; (**c**) ZUE/DM; (**d**) XUE/DM; (**e**) 2UE/DM.

**Figure 10 materials-15-00617-f010:**
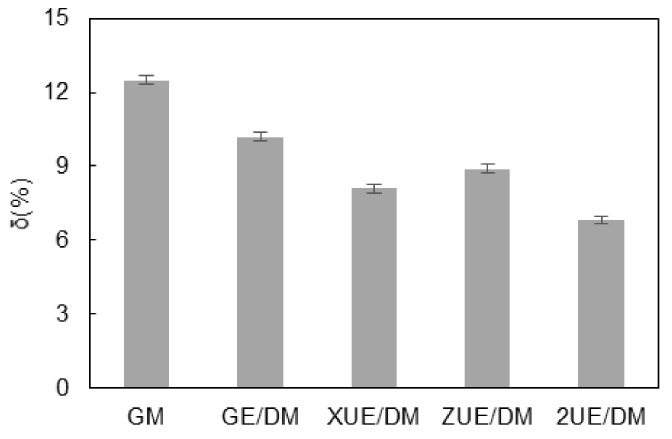
Comparison of form precision.

**Figure 11 materials-15-00617-f011:**
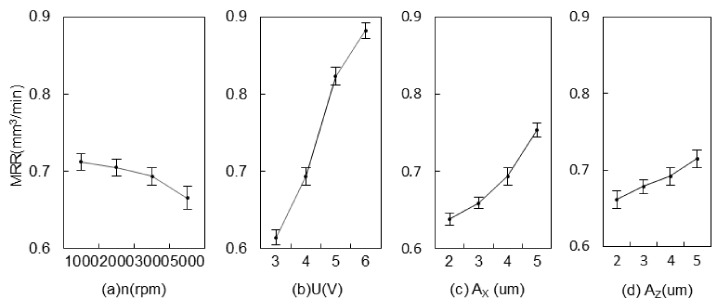
Effects of machining parameters on MRR.

**Figure 12 materials-15-00617-f012:**
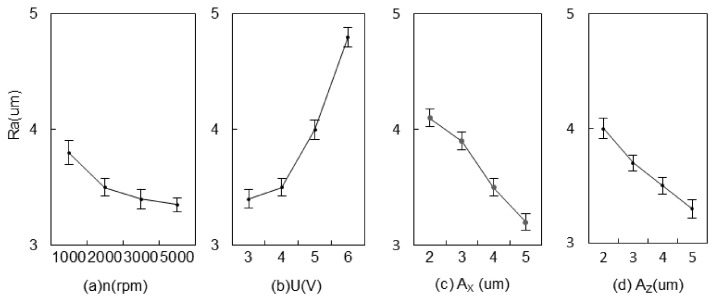
Effects of machining parameters on Ra.

**Figure 13 materials-15-00617-f013:**
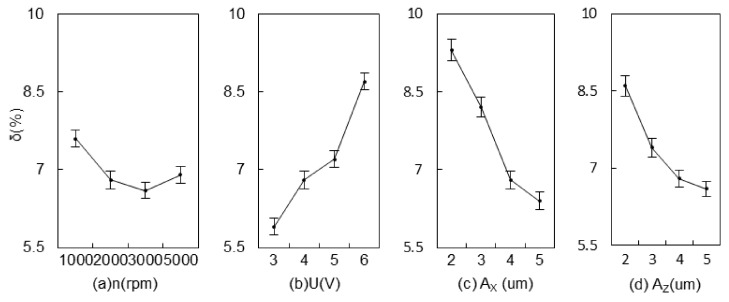
Effects of machining parameters on δ.

**Figure 14 materials-15-00617-f014:**
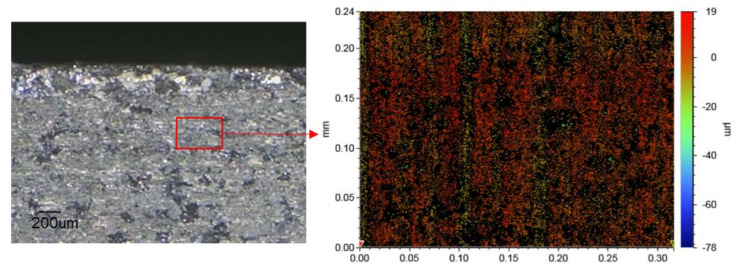
The machined surface and topography in optimized parameters.

**Table 1 materials-15-00617-t001:** Machining properties of 40 SiCp/Al.

Material	APS (μm)	Density (g/cm^3^)	Yield Strength (MPa)	Modulus of Elasticity (GPa)
**6061Al 40vol.%**	10	2.9	210	163

**Table 2 materials-15-00617-t002:** Machining conditions for the mechanism.

Category	GM	GE/DM	XUE/DM	ZUE/DM	2UE/DM
Static pressure	5N				
Electrolyte	0.5 wt. % NaNO_3_				
Spindle speed (rpm)	3000				
Feed speed (mm/min)	12				
Machining depth (mm)	0.01				
Applied voltage (V)	0	4	4	4	4
X amplitude (μm)	0	0	4	0	4
Z amplitude (μm)	0	0	0	4	4

**Table 3 materials-15-00617-t003:** Machining parameters for 2UE/DM.

Group	Spindle Speed: n (rpm)	Applied Voltage: U (V)	X Amplitude: A_X_ (μm)	Z-Amplitude: A_Z_ (μm)
1	1000, 2000, 3000, 5000	4	4	4
2	3000	3, 4, 5, 6	4	4
3	3000	4	2, 3, 4, 5	4
4	3000	4	4	2, 3, 4, 5

**Table 4 materials-15-00617-t004:** Level of factors for 2UE/DM.

Levels	A:n (rpm)	B:U (V)	C:Ax (μm)	D:Az (μm)
**−1**	1000	3	2	2
**1**	5000	6	5	5

**Table 5 materials-15-00617-t005:** Results of the orthogonal analysis.

Run No.	A	B	C	D	MRR (mm^3^/min)	Ra (μm)	δ (%)
**1**	−1	−1	−1	−1	0.68	4.28	8.43
**2**	−1	−1	1	1	0.77	3.43	5.96
**3**	−1	1	−1	1	0.87	4.52	9.27
**4**	−1	1	1	−1	0.90	4.35	8.82
**5**	1	−1	−1	1	0.65	3.79	7.32
**6**	1	−1	1	−1	0.72	3.60	7.23
**7**	1	1	−1	−1	0.79	4.61	10.42
**8**	1	1	1	1	0.91	3.78	8.21

**Table 6 materials-15-00617-t006:** ANOVA for MRR, Ra, and δ.

Source	F (MRR)	P (MRR)	F (Ra)	P (Ra)	F (δ)	P (δ)
**A**	21.43	0.036	9.83	0.004	0.02	0.197
**B**	77.36	0.001	28.11	0.000	106.36	0.000
**C**	17.36	0.005	44.49	0.000	12.19	0.001
**D**	13.71	0.076	21.33	0.001	14.82	0.002

## Data Availability

Data is contained within the article.

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
