# Peer review of "Study on Generating Machining Performance of Two-Dimensional Ultrasonic Vibration-Composited Electrolysis/Electro-Discharge Technology for MMCs"

_materials, 2022, doi:10.3390/ma15020617_

Round 1
Reviewer 1 Report
Literature related to grinding and EDM needs to be added.
Regarding the shape precision Calculation, did you get it from literature, please provide the reference. How did you calculate SB.
Explain in details figure 5(what is DM)
Is it correct (The electrolysis currents of both 2UE/DM and ZUE/DM vary with the periodic change in the gap) as per the figure 5? Figure 5b shows different trend with respect figure 5d.
Check the cation of figure 5a.
Check reference 9
Provide the materials property of workpiece.
What are the cutting condition of general grinding machining?
What are the process parameters of without vibration (GE/DM)?
What is the frequency of vibration used in the experiment?
Table 1 does not provide spindle speed, machining depth or feed for GM? Was it dry grinding or wet, if wet what kind of cutting fluid was used and how did you apply cutting fluid.
How did you compare the performance parameters for 4 different kinds of machining, what are the common basis to compare when they have different process parameters?
Some typo, and sentence formation issue.
Reviewer 2 Report
1) Kindly please enhance the language standard
2) Abstract should be refined
3) More recent papers related to application of ultrasonics to EDM /ECM process published last 5 years; You may use follows; https://www.tandfonline.com/doi/abs/10.1080/10426914.2021.2001524?journalCode=lmmp20; Machining behaviors of vibration-assisted electrical arc machining of W9Mo3Cr4V; Multi Criteria Decision Making of Vibration Assisted EDM Process Parameters on Machining Silicon Steel Using Taguchi-DEAR Methodology
4) Novelty can be provided seperately
5) Surface roughness can be denoted as 'Ra' instead of 'SR'
6) 'Feed speed' may be termed as 'Feed rate'
7) Kindly please refine conclusion section
Best wishes and looking for revised version
Reviewer 3 Report
The manuscript analysed the material removal mechanism and the main performances (tool wear rate and surface roughness) of different processes for particulate-reinforced aluminium alloys. The manuscript is interesting for its contents but there are some aspects that must to be fixed before proceeding with the publication.
First of all, the goal of the research is not clearly described in the introduction. Please provide a clarification about that.
In section 3 the authors described the experimental set-up. The characteristics of the equipment are depicted in a very specific way, but the selected factorial plan is not clear. In particular:
- How have you selected the levels of machining parameters?
- Did you perform several runs or a single test for each combination of machining parameters?
- In each group, if I understand well, you performed 4 tests with different combination of machining parameters. Why these groups do not appear in the bar cart of the performances level? Did you use the average value of all these tests for the value of 2UE/DM reported in Fig 6 and Fig.7?
In section 5 the authors describe the effects of the variable parameters on the main process performances. The description is simply related to the observation of the trend of the curve in the plots reported in Fig. 11, 12 and 13. This means that the authors cannot be sure about the effects of the parameters in the indicators, since they do not perform any statistical analysis that allows to support these considerations. It is simply the description of what is represented in the plots. I suggest to applied an analysis of variance to MRR, Ra and shape precision for identifying which are the parameters that actually generate an effect on the performances.
In figure 11, 12 and 13 are reported the symbols for the parameters and in soma cases they have not been introduced before. Please introduce the sign of each factor maybe in table 2.
Round 2
Reviewer 1 Report
Accepted in the present form.
Author Response
Dear reviewers,
Thank you very much for you appreciation.The english language has been modified in detail.
Reviewer 2 Report
The manuscript may be accepted in its present form.
Author Response

(The authors gave the same response as above.)

Reviewer 3 Report
The authors respond satisfactorily to the previous comments improving the level of the manuscript.
Author Response

(The authors gave the same response as above.)
